# Latin American Genes: The Great Forgotten in Rheumatoid Arthritis

**DOI:** 10.3390/jpm10040196

**Published:** 2020-10-26

**Authors:** Roberto Díaz-Peña, Luis A. Quiñones, Patricia Castro-Santos, Josefina Durán, Alejandro Lucia

**Affiliations:** 1Faculty of Health Sciences, Universidad Autónoma de Chile, Talca 3460000, Chile; 2Laboratory of Chemical Carcinogenesis and Pharmacogenetics, Department of Basic-Clinical Oncology, Faculty of Medicine, University of Chile, Santiago 8320000, Chile; 3Latin American Network for Implementation and Validation of Clinical Pharmacogenomics Guidelines (RELIVAF-CYTED), 28015 Madrid, Spain; 4Inmunología, Centro de Investigaciones Biomédicas (CINBIO), Universidad de Vigo, 36310 Vigo, Spain; patricassan@gmail.com; 5Department of Rheumatology, School of Medicine, Pontificia Universidad Católica de Chile, Santiago 7690000, Chile; jgduran@uc.cl; 6Faculty of Sport Sciences, European University of Madrid, 28670 Madrid, Spain; alejandro.lucia@universidadeuropea.es; 7Research Institute Hospital 12 de Octubre (imas12), 28041 Madrid, Spain

**Keywords:** rheumatoid arthritis, genetics, genome-wide association studies, polymorphism, ethnic group, admixed population

## Abstract

The successful implementation of personalized medicine will rely on the integration of information obtained at the level of populations with the specific biological, genetic, and clinical characteristics of an individual. However, because genome-wide association studies tend to focus on populations of European descent, there is a wide gap to bridge between Caucasian and non-Caucasian populations before personalized medicine can be fully implemented, and rheumatoid arthritis (RA) is not an exception. In this review, we discuss advances in our understanding of genetic determinants of RA risk among global populations, with a focus on the Latin American population. Geographically restricted genetic diversity may have important implications for health and disease that will remain unknown until genetic association studies have been extended to include Latin American and other currently under-represented ancestries. The next few years will witness many breakthroughs in personalized medicine, including applications for common diseases and risk stratification instruments for targeted prevention/intervention strategies. Not all of these applications may be extrapolated from the Caucasian experience to Latin American or other under-represented populations.

## 1. Introduction

Rheumatoid arthritis (RA) is a multifactorial, inflammatory, and progressive autoimmune disease that affects approximately 1% of the population worldwide [1]. As in other complex systemic diseases, the global variations in the prevalence, clinical manifestation, and prognosis of RA and its risk factors are driven by environmental, socioeconomic, demographic, cultural, and genetic factors [1]. The majority of studies on genetic and environmental risk factors for RA have been performed in populations of European descent, estimating a prevalence of 0.5–1.1% for this condition [1]. One study examining clinical manifestations of RA in different ethnic groups found that clinical manifestations and shared epitope (SE) expression varied, although clinical expression might have been influenced by socioeconomical differences [2]. Patients with RA are not a homogenous population and some clinical RA subgroups, such as anti-citrullinated protein autoantibodies (ACPA) seropositive versus seronegative, erosive versus non-erosive, and progressive versus mild course, have been identified [3].

The implementation of personalized medicine is based on the processing of information obtained at a population level and its subsequent integration with the specific biological, genetic, and clinical characteristics of an individual. Other factors that can provide additional complexity include ethnicity, lifestyle, and environmental factors, as well as co-existing susceptibility factors for other diseases. In this context, there is a relevant gap to bridge between Caucasian and non-Caucasian target populations, with the latter under-represented in these types of analyses. Indeed, without the extension of genetic studies to other populations, the transferability of personalized medicine to non-Caucasian populations might not be feasible [4]. The development of omics-based research and medicine has brought considerable progress in our understanding of complex diseases, including the rheumatic diseases. In this review, we summarize our current understanding of genetic determinants of RA risk in global populations, but with a focus on Latin American populations. The next few years will likely witness the widespread use of personalized genomics, with applications for common diseases and risk stratification instruments for targeted prevention/intervention strategies. Not all of these applications may be extrapolated from the Caucasian experience to Latin American populations.

## 2. Rheumatoid Arthritis in Latin American Populations

The relevance of genetics in the pathogenesis of RA is particularly evident in Latin Americans [5], as female predominance is greater than what has been classically described in Caucasian populations: the female-to-male ratio in Latin American populations is 6:1 compared with 3:1 in Caucasians [6]. With regard to prevalence in Latin American populations, studies have reported values ranging from 0.3% to almost 2%. For instance, in Mexico, the prevalence was found to be 1.6%, but with values ranging from 0.9% to 2.8% in different regions [7]. In Argentina, one study in the northwest found a prevalence of 1.9% (95% confidence interval: 1.8–2.0), whereas two studies performed in the central region found the prevalence to be less than 1% [8,9]. A study in Colombia found a prevalence greater than 1%, as in the Mexican study, whereas in Cuenca (Ecuador), Tambo Viejo (Perú), Montes Claros (Brazil) and Monagas (Venezuela), the prevalence of RA was closer to 0.5%, with point estimators of 0.8% (0.5–1.2), 0.5% (0.19–0.82), and 0.4% (0.2–0.6), respectively [10,11,12,13,14]. Finally, in a population-based study in Chile, the prevalence of RA was 0.6% (0.3–1.2) [15]. Although many of these studies have methodological limitations, it seems that the prevalence of RA is not uniform in Latin America and this is likely related to a variable genetic background in local indigenous populations as well as to the influence of European ancestry, which varies in different areas of Latin America.

## 3. Genetics in Rheumatoid Arthritis

Most rheumatic diseases are multifactorial, with genetic variability, environmental factors, and random events interacting to modulate pathological pathways [16]. Genetic susceptibility in RA is evident in familial clustering and monozygotic twin studies [17]. Approximately 50% of RA risk is thought to be genetic, with the strongest associations linked to the human leukocyte antigen (HLA) region, and heritability of RA has been estimated to be ~60% [17]. Disease progression, outcome, and RA phenotype have also been associated with genetic factors [18,19]. While the identification of the predisposing genetic polymorphisms in RA provides clues to the understanding of the mechanisms involved in its etiopathogenesis, there are major challenges in the interpretation of the clinical relevance of the genetic variants associated with disease and the identification of their involvement in the different clinical manifestations. Overcoming this hurdle will make it possible to predict the evolution of the disease, to establish new treatments, and to address the development of personalized therapies. Furthermore, the potential to identify individuals at significantly higher genetic risk will present a number of opportunities and challenges for clinical medicine.

### 3.1. Human Leukocyte Antigen Region

The human “major histocompatibility complex” (MHC) region, HLA, located on chromosome 6 (Figure 1), is by far the strongest source of genetic susceptibility to RA. The region consists of genes encoding molecules with a key role in the immune system by modulating responses to invading pathogens. Approximately one-third of genetic risk is associated with the HLA locus [17]. Its strong association dates back to 1978 [20], specifically to the SE of HLA-DRB1, which encodes a common amino acid sequence associated with RA susceptibility and progression (^70^QRRAA^74^, ^70^QKRAA^74^, or ^70^RRRAA^74^) [21]. HLA-DR loci belong to the HLA class II molecules (DR, DQ, and DP) that function as receptors for processed peptides derived predominantly from membrane and extracellular proteins. HLA-DR expression is located on the surface of antigen-presenting cells (APC) (including dendritic cells, macrophages, and B cells) and is essential to display peptides to CD4+ T cells, inducing their activation. The presence of an SE suggests that the HLA alleles bind the same antigen, postulating the presentation of arthritogenic self-peptides or molecular mimicry with foreign antigens and/or shaping the T-cell-antigen repertoire. However, this arthritogenic peptide has not yet been identified.

*HLA-DR* complex is highly polymorphic, mainly in the DRβ1 chain encoded by the *HLA-DRB1* gene [22]. To date, 2838 *HLA-DRB1* alleles encoding 1973 proteins have been identified (http://hla.alleles.org/nomenclature/stats.html, last accessed on 24 October 2020). The exon 2 of *HLA-DRB1* is the most variable one and is responsible for the localization of the antigen recognition site. Thus, differences in antigenic presentation can be largely interpreted in terms of the effect of polymorphisms in *HLA-DRB1*. There are ethnic differences in the presence of specific *HLA-DRB1* SE alleles worldwide (Table 1) [23]. *HLA-DRB1* SE alleles are the most important genetic contributors to the risk of developing ACPA-positive RA, particularly in Caucasians [24]. However, it is unclear whether this association is valid in all ethnic groups. Some *HLA-DRB1* alleles have been associated with RA in Native Americans, individuals with Mexican American ancestry, as well as in Chilean, Peruvian, Colombian, Brazilian, and Mexican Mestizo populations with a larger proportion of European ancestry [25]. A meta-analysis confirmed a significant association between RA and the *HLA-DRB1* gene and revalidated the SE hypothesis in Latin American populations [25].

López Herráez et al. genotyped 196,524 single-nucleotide polymorphism (SNP) markers in 1475 patients with RA and 1213 controls [26] and revealed the complex genetic architecture of the HLA region related to RA in Latin American populations. Samples were obtained from different Latin American countries, including Argentina, Mexico, and Peru, and also included 135 patients and 78 controls from a Chilean population. A strong association with HLA region was observed, with three independent effects, probably due to the diverse origin of the patients and disparities in the sample size among populations. Accordingly, efforts are needed to clarify the role of HLA in RA and the differences in Latin American populations. It is very relevant that the imputation of HLA alleles in Latin American populations may be controversial due to the scarcity of available information and the complexity of the admixture. Currently, classic HLA typing is clearly the best tool, despite its high cost.

### 3.2. Genome-Wide Association Studies

Our understanding of genetic susceptibility to disease has greatly improved by the ever-increasing power of genome-wide association studies (GWAS) based on SNP arrays. Since 2005, around 50,000 risk loci and their proposed functions have emerged from almost 4000 GWAS and GWAS meta-analyses [27].

#### 3.2.1. Non-HLA Genetic Associations

To date, over 100 genetic loci have been associated with RA [28]. The pathogenesis of RA has a polygenic basis, where 50% of disease risk is thought to be genetic and one-third of this risk is attributable to the HLA region [17]. Thus, genetic variation can also be explained by RA risk alleles in non-HLA loci. In early GWAS, peptidyl arginine deiminase 4 (*PADI4*), protein tyrosine phosphatase non-receptor type 22 (*PTPN22*), and cytotoxic T-lymphocyte-associated protein 4 (*CTLA4*) genes were identified as initial non-HLA RA risk loci [29,30] (Figure 2), with these findings being subsequently replicated in multiple cohorts in the following years [28]. There has been an exponential increase in the number of genes associated with RA since 2007, as several collaborative efforts have led to the development of international consortia [31]. In total, over 90 non-HLA RA risk loci have emerged from GWAS and subsequent meta-analyses of GWAS datasets. Among the coding variants showing a strong association with RA risk are *PTPN22*, interleukin-6 receptor (*IL6R*), and tyrosine kinase 2 (*TYK2*) [32]. However, around 80% of the loci are located in non-coding regions of the genome [28], including alleles found to lie either in intronic regions or in proximal 5′ or 3′ intergenic regions of immune-mediating genes, notably interferon regulatory factor 5 (*IRF5*) and 8 (*IRF8*), runt-related transcription factor 1 (*RUNX1*), TNF alpha induced protein 3 (*TNFAIP3*), signal transducer and activator of transcription 4 (*STAT4*), chemokine receptor 6 (*CCR6*), and *CTLA4* [28].

#### 3.2.2. Ancestral Diversity and Sample Sizes

While the success of GWAS is undeniable [33], the majority of the populations studied in GWAS are of European descent. Participants are typically ~80% European descent and <1% Latin American ancestry [27]. Moreover, cohorts included in some GWAS are characterized repeatedly. As sample sizes for Caucasian in GWAS continue to grow, those for other populations stagnate. Indeed, sample sizes of over 1 million participants have been reached for some phenotypes [34], with genotype and clinical information and even sequencing data. Overall, this would suggest a paucity of transferable genetic findings across different populations.

With regard to GWAS of the rheumatic diseases, there is also a European bias. Figure 3A shows the proportion of individuals with different ancestries represented in GWAS performed for RA, systemic lupus erythematosus, osteoarthritis, ankylosing spondylitis, psoriatic arthritis, and Behçet’s disease; we included a total of 4194,198 individuals extracted from the GWAS Diversity Monitor [35]. The results reveal that ancestry in genetic studies is dominated by participants of European ancestry (89.60% combined) and is unequal in general terms. Moreover, by dividing these data into discovery and replication samples, the percentage of European ancestry samples used for the initial phase is significantly higher than for replication studies (94.57% for discovery and 78.22% for replication; *p*-value < 10^−5^). In terms of participants of non-European ancestry, only Asians have a significant representation (4.45% discovery, 20.03% replication, 9.18% combined). In contrast to what is observed in Caucasians, however, the percentage of Asian ancestry samples used for the initial analysis is significantly lower than that for replication studies (*p*-value < 10^−5^). Other studied ancestries include African American or Afro-Caribbean (0.31% discovery, 0.06% replication, 0.23% combined), African (0.03% discovery, 0.00% replication, 0.02% combined), Hispanic or Latin American (0.16% discovery, 0.01% replication, 0.11% combined), and Other or Mixed (0.48% discovery, 1.69% replication, 0.85% combined). Focusing on RA (Figure 3B), the data are very similar, with greater involvement of participants of Asian ancestry, to the detriment of those of European ancestry. What does appear clear is that for the non-European ancestry participants, their inclusion depends on the involvement of individual researchers in international consortia at the discovery stage. The usual procedure for GWAS is firstly a discovery stage in Caucasians, followed by a replication stage in Caucasian and Asian cohorts, and the data demonstrate the under-represented contribution of associations from African and Hispanic or Latin American ancestries when compared with the total number of individuals included in GWAS.

#### 3.2.3. Population-Specific and Rare Variants

The small proportion of Hispanic or Latin American participants in GWAS suggests great opportunities for new discoveries in these populations [4]. Latin Americans have been designated as “Hispanics” and considered homogenous in most of the GWAS where they have been included. However, the degree of admixture varies between Latin American countries according to the major ancestry component [36]. In fact, their genomes are a mosaic of different ancestral origins, as the beginnings and destinations of these populations have depended on the time and reasons for their migration. Admixture mapping, an approach to studying the association between local genetic ancestry and disease risk in the genome, may also facilitate the discovery of loci associated with traits [37].

It is increasingly apparent that genomic diversity among populations can provide new opportunities for discovery, and this is particularly evident in the case of population-specific variants. For example, Bergström et al. recently investigated the extremes of human genetic variation by detecting variants that are “private” (found only in a single population) to geographic regions [38]. They found an excess of previously undocumented common genetic variation private to the Americas, Southern Africa, Central Africa, and Oceania and absent in other geographical regions (Europe, East Asia, the Middle East, or Central and South Asia). Each of these populations harbors tens to hundreds of private genetic variants, most arising as new mutations. To get an idea of the huge variability and particularity, even comparing Central and South America, the authors found variants private to one region but absent from the other region reaching >40% frequency [38]. Geographically restricted genetic variation is likely to have biomedical implications but will remain unknown until genetic association studies are extended to include Latin Americans and other currently under-represented ancestries.

Population-specific variants are more likely to be rare variants and recent in origin, in relation to common ancestral variants. Rare or low-frequency variants could explain the substantial unexplained heritability of many complex diseases (i.e., “missing heritability”), most of which are not well captured by current genotyping technology since the majority of SNPs used in GWAS are common variants. This is exacerbated when under-represented populations are surveyed, because most arrays commonly provide poor coverage in non-Caucasian populations and genotype imputation cannot be performed. Moreover, the effect sizes are likely to be larger in rare than in common variants [39]. Rare variants are expected to be under purifying selection and thus enriched with deleterious, protein-coding mutations participating in complex traits [40]. Next-generation sequencing technologies and whole exome/genome sequencing efforts are ongoing in many populations worldwide [38,41] and will likely become the engine to uncover rare variants that are specific to an ethnic group or population and to study their association with complex phenotypes.

There have been few published reports identifying rare variants in RA. In 2010, Bowes et al. applied a rare variant collapsing method to a large GWAS dataset [42], prioritizing gene regions for further investigation in an independent cohort of RA cases and controls. The authors were able to detect replicating association to low-frequency variants in tumor necrosis factor, alpha-induced protein 3 (*TNFAIP3*), a known RA risk gene. In another study, Bang et al. analyzed targeted exon sequencing data of 398 genes selected from a multifaceted approach in a Korean population [43], but they were unable to identify rare coding variants with large effect to explain the missing heritability for RA. In a study by Diogo et al. on the deep exon sequencing of 25 candidate genes from GWAS [44], the authors combined DNA in 10 pools of 50 patients with RA and 13 pools of 50 matched controls (each pool containing the same amount of DNA from each individual), identifying an accumulation of missense variants in the interleukin 2 receptor subunit alpha (*IL2RA*) and beta (*IL2RB*) gene, and showing that variants within the protein-coding portion of a subset of biological candidate genes identified by GWAS may contribute to RA risk. More recently, Li et al. carried out a whole-exome sequencing study in 58 patients with RA and 66 healthy controls in the Han Chinese population [45], identifying genes enriched with deleterious variants that were involved in innate immunity pathways and contributed to the risk of RA. Thus, rare coding variants across the whole genome could participate in the missing genetic contribution to RA.

We recently performed high-density SNP genotyping in candidate genes to test their association with susceptibility to RA in the Chilean population [46], in an attempt to provide insight into the cross-ethnic generalizability of known Caucasian and Asian RA risk loci to Latin American populations. We showed that allele frequency varies between populations of different ancestries (Figure 4A), suggesting the existence of genomic patterns in Chilean and, likely, other Latin American populations, which differentiate them from Caucasian with regard to loci that are relevant for RA (Figure 4B).

#### 3.2.4. Implications for Variant Discovery and Medical Genetics

Genetic variation at the population level is itself shaped by population history and demography. While GWAS cannot identify 100% of the heritability of complex traits, nearly every GWAS has, in our opinion, overlooked the geographical component. Gene–environment interactions are expected to explain a portion of the missing heritability [47], but their impact has not been investigated in any great detail. Genetic susceptibility to disease should be analyzed in the context of environmental risk factors that have a direct relationship. Different subsets of genes could play relevant roles depending on the risk environment. For instance, ultraviolet radiation is a known risk factor for rheumatic diseases. It seems intuitive that individuals with high genetic predisposition would have an increased risk, for example, if they resided in areas with higher sunlight exposure. The expression of genetic variants is modified by many environmental factors, and the significance of ethnicity in genetics is a factor to take into account [48]. What is clear is that populations have distinct histories and social systems, and the relationship with the exposure to certain disease factors or traits is a reversible reaction.

The effects of individual SNP markers are limited, but collectively, they provide meaningful insights into underlying pathways and can contribute to models for risk-stratification for some common diseases, such as RA [49]. Recently, Khera et al. proposed that now is the time to contemplate the inclusion of polygenic risk prediction or polygenic risk scores (PRS) in clinical care [50]. They describe scores for common diseases (coronary artery disease, atrial fibrillation, type 2 diabetes, inflammatory bowel disease, and breast cancer) that identify individuals with risk equivalent to monogenic mutations. We are confronted here once more with the fact that the PRS described were derived and tested in individuals of primarily European ancestry. It is necessary to ensure that all ethnic groups have access to genetic risk prediction, which will require undertaking or expanding GWAS in non-Caucasian ethnic groups. Otherwise, the clinical use of PRS may increase health disparities [51].

## 4. Pharmacogenetics and (Pharmaco) Epigenomics

### 4.1. Pharmacogenetics

Pharmacogenetics is the study of variations in genes encoding drug transporters, drug-metabolizing enzymes, and drug targets and their translation to differential responses to drugs. Although the use of genetic markers to predict treatment response in RA patients has huge potential and pharmacogenetic variability can be reliably determined at a relatively low cost, so far, pharmacogenetic studies have failed to find reliable associations [52]. The ultimate goal of pharmacogenetics in rheumatology is to define genetically distinct subsets of patients who have differential responses to the various therapies used to treat rheumatic diseases. In this regard, the antifolate agent methotrexate (MTX) is the first-line disease-modifying agent for the treatment of RA worldwide, but patients show large variability in response to this drug, with some particularly exposed to adverse effects. Indeed, approximately 30% of patients experience inadequate treatment response and around 20% stop MTX due to toxicity [53].

Despite the inconsistent results, pharmacogenetics might represent a useful means for optimizing MTX therapy in patients with RA. In fact, inconsistencies might not be surprising. Response to treatment in RA can be measured using different approaches. One of the most common methods is the change in Disease Activity Score of 28 joints (DAS28), which includes subjective patient-reported outcome measures, such as the tender joint count (TJC) and the visual analog scale (VAS) [54,55]. Moreover, variability in initial, escalation, and maintenance dosage of MTX, or use of concomitant disease modifying anti-rheumatic drugs (DMARDs), has not been accounted for in previous genetic studies of response to MTX. Biologics include anti-tumor necrosis factor (TNF)α drug therapy (such as infliximab, adalimumab, etanercept, golimumab, and certolizumab), B-cell depleters (rituximab), IL-1 (anakinra) and IL-6 inhibitors (tocilizumab), and inhibitors of T-cell co-stimulation (abatacept). TNFα is a pleiotropic inflammatory cytokine, and its expression plays a central role in the pathogenesis of autoimmune diseases. Anti-TNFα agents are widely used worldwide. Correct stratification of patients is an important issue, because individuals show large variability in response to this type of treatment and, as such, some are exposed to adverse effects. In fact, although anti-TNFα therapy is effective in the majority of cases, 30 to 40% of treated patients show an inadequate response [56]. Biological therapies are still prescribed on a “trial-and-error” basis, where a delay in achieving control of inflammation results in worse long-term outcomes. At present, although there are some candidates [57], no genetic biomarker for anti-TNFα therapy has been confirmed in large cohorts of patients or in meta-analyses. There are still major challenges when investigating the pharmacogenetics of treatment response in RA. Large studies with integrative analyses are likely to play a key role in future studies aiming at identifying disease biomarkers. As in other aspects discussed in this review, there are no published data available about the pharmacogenetics of RA in Latin American populations.

### 4.2. Epigenomics

A high number of environmental factors, including ultraviolet (UV) radiation, vitamin D levels, specific infections, or the patient microbiome, can be involved in the pathogenesis of rheumatic diseases [16]. However, the nature of the environmental component is poorly understood. Environmental factors can influence susceptibility to these conditions through their influence on epigenetic modifications [58].

Epigenetics comprises all the heritable changes in gene activity that do not involve an alteration in the primary DNA sequence. Epigenetic modifications affect the condensation rate of DNA and therefore its accessibility for ribonucleic acid (RNA) transcription machinery, thereby impacting gene expression. Although epigenetic modifications are heritable, they are also dynamic, being usually modified by environmental factors. Epigenetic phenomena include DNA methylation, post-translational histone modifications, and microRNAs (miRNA). The study of the epigenetic characteristics of patients with RA opens a new field for identifying novel biomarkers and therapeutic targets as well as individual variability in response to a specific treatment.

Neutrophils, B and T cells, macrophages, osteoclasts, and synovial fibroblasts show an activated response in the joints of patients with RA. The activation of these cells induces the secretion of enzymes and other molecules that contribute to the destruction of cartilage and bone tissues. Several studies on epigenetics in RA patients have reported some specific changes in DNA methylation (Table 2).

In general, and in contrast with most cancers (in which hypermethylation is rather widespread), candidate genes tend to be hypomethylated in the context of RA. Other epigenetic changes, histone modifications, and abnormal expression of non-coding RNAs can also contribute to RA development [68].

An important issue in epigenetic studies is that the relevant modifications may be only present in a particular tissue or cell type. Therefore, only the damaged tissue epigenome is relevant, and studies should focus solely on those in cells that are actually involved in the pathogenesis of the disease. Another relevant issue to take into consideration is whether the epigenetic modifications associated with disease development are actually a cause or a consequence of the inflammatory process.

One of the mechanisms of action of some treatments relies on the specific drug effect on epigenetic modifications. Thus, as with some anti-cancer therapies, a previous determination of the epigenetic modifications might be needed prior to personalized treatment to ensure an optimal response. In this regard, DNA methylation is stable over time and does not fluctuate, as opposed to protein or mRNA levels. Therefore, DNA methylation status could be theoretically used a robust biomarker if we could determine an association with the response to therapy. This possibility, however, might not apply to histone modifications because they change more quickly over time [69]. The methylation status prior to the treatment in question can provide relevant information for the prediction of an individual’s response to a specific therapy. TNFα inhibitors used to treat rheumatic diseases might act by altering and reversing causative epigenetic changes underlying chronic disease [70,71].

Unlike genetic lesions, epigenetic alterations are reversible and could be modulated by diet, drugs, and other environmental factors. This epigenetic flexibility suggests the use of strategies for the prevention and treatment of diseases where epigenetic factors are known to play a pathogenic role. However, most epigenetic studies to date are underpowered because they were performed in racially homogeneous populations, which limits their generalizability to other ethnic groups, notably Latin American populations. Large, prospective, and multicenter studies, which are also multi-ethnic, are needed to overcome this and other problems of reproducibility.

## 5. Conclusions

Advances in GWAS have greatly improved our understanding of pathogenic mechanisms underlying autoimmune diseases in general and RA in particular [32,72,73]. Moreover, genetic risk scores have been developed for both prediction of RA progression and for susceptibility [74,75,76]. Recently, Knevel et al. showed that genetic information can add value to the clinical information in rheumatology [77]—specifically, with contributions to discriminate between five different inflammatory arthritides (RA, systemic lupus erythematous, spondyloarthropathies, psoriatic arthritis, and gout). The algorithm developed, termed G-PROB, could show greater diagnostic accuracy than a rheumatologist’s predictions at a patient’s first visit. Overall, genetic risk scores have had modest predictive value in RA, but symptom-based selection seems to provide an opportunity for pre-testing for disease. Polygenic risk scores are of potential use in rheumatology, but further research is required. A focus on the identification of causal variants and the characterization of their functional consequences is perhaps now a more important area of study. Combining this with progress in the area of pharmacogenomics could lead to great advances in the management of RA. However, the existing gap between Caucasian and non-Caucasian target populations, and above all in relation to the African and Hispanic or Latin American populations, appears insurmountable at the moment. This lack of ethnic diversity will substantially decrease the capacity for social benefit [78]. Genomic diversity among populations offers new opportunities for discovering genetic variants in RA that may provide keys to understanding the pathogenic mechanisms involved and may be important for the prediction of disease severity, prognosis, and response to treatment.

Currently, progress in our understanding of the genetics of pathogenic mechanisms underlying RA in Latin American countries depends, for the most part, on the inclusion of individual researchers in international consortia. Researchers in Latin American countries often start projects with few resources, since the national funding agencies provide little help, especially in certain topics. Without neglecting a collaborative mindset, we require a change in the funding strategy in order to address the needs of understudied populations and to move towards a more complete picture of the genetic architecture of complex traits across populations.

## Figures and Tables

**Figure 1 jpm-10-00196-f001:**
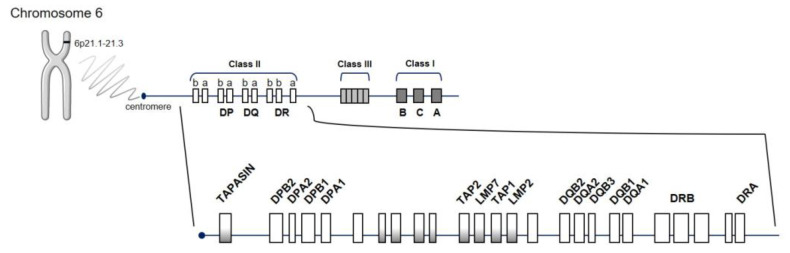
Schematic representation of the human leucocyte antigen (HLA) region. HLA molecules are encoded by genes located on short arm of chromosome 6 at position 6p21.1–21.3, in class I and class II regions. Abbreviations for the different genes located in the HLA complex: DPA1 and DPA2: DP alpha 1 and 2 chain, respectively; DPB1 and DPB2: DP beta 1 and 2 chain, respectively; DRA, DR alpha chain; DRB, DR beta chain; DQA1 and DQA2: DQ alpha 1 and 2 chain, respectively; DQB1 and DQB2: DQ beta 1 and 2 chain, respectively; LMP2 and LMP7, latent membrane protein 2 and 7, respectively; TAP1, transporter associated with antigen processing 1; TAPASIN, TAP-associated glycoprotein (also known as TAPBP).

**Figure 2 jpm-10-00196-f002:**
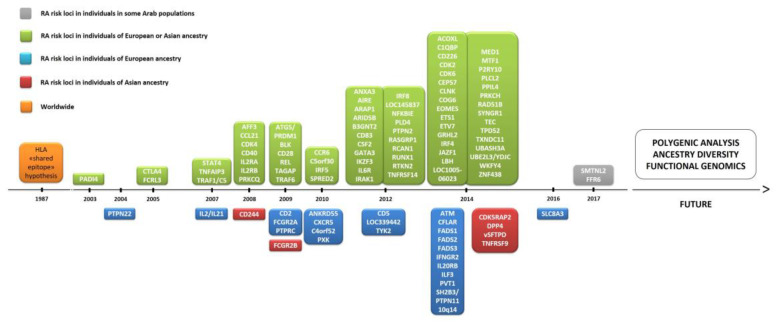
Timeline of the discovery of several genes associated with rheumatoid arthritis (RA). Abbreviations: ACOXL, acyl-CoA oxidase like; AFF3, AF4/FMR2 family member 3; AIRE, autoimmune regulator; ANKRD55, ankyrin repeat domain 55; ANXA3, annexin A3; ARAP1, ArfGAP with RhoGAP domain, ankyrin repeat and PH domain 1; ARID5B, AT-rich interaction domain 5B); ATM, ATM serine/threonine kinase; B3GNT2, UDP-GlcNAc:betaGal beta-1,3-N-acetylglucosaminyltransferase 2; C1QBP, complement C1q binding protein; C4orf52, chromosome 4 open reading frame 52; C5orf30, chromosome 5 open reading frame 30 (also known as MACIR (macrophage immunometabolism regulator)); CCL21, chemokine (C-C motif) ligand 21; CCR6, chemokine receptor 6; CD2, CD226, CD244, CD40, CD5 and CD83: cluster of differentiation 2, 226, 40, 5, and 83, respectively; CDK2, CDK4 and CDK6: cyclin dependent kinase 2, 4, and 6, respectively; CDK5RAP2, CDK5 regulatory subunit-associated protein 2; CEP57, centrosomal protein 57; CFLAR, CASP8 and FADD like apoptosis regulator; CLNK, cytokine-dependent hematopoietic cell linker; COG6, component of oligomeric Golgi complex 6; CSF2, colony stimulating factor 2; CTLA1, cytotoxic T-lymphocyte antigen 1; CXCR5, chemokine receptor type 5; DPP4, dipeptidyl-peptidase 4; EOMES, eomesodermin (also known as T-box brain protein 2 (Tbr2)); ETS1, ETS proto-oncogene 1, transcription factor; ETV7, ETS variant transcription factor 7; FADS1, FADS2, and FADS3: fatty acid desaturase 1, 2, and 3, respectively; FCGR2A, Fc fragment of IgG receptor IIa; FCG2RB, Fc fragment of IgG receptor IIb; FCRL3, Fc receptor-like protein 3; GGT6, gamma-glutamyltransferase 6; GATA3, GATA binding protein 3; GRHL2, grainyhead-like transcription factor 2; HLA, human leukocyte antigen; IFNGR2, interferon gamma receptor 2; IL2 and IL21: interleukin 2 and 21, respectively; IL2RA, interleukin-2 receptor alpha chain; IL2RB, interleukin 2 receptor subunit beta; IL6R, interleukin 6 receptor (also known as CD126 (cluster of differentiation 126)); IL20RB, interleukin-20 receptor beta chain; ILF3, interleukin enhancer-binding factor 3; IKZF3, zinc finger protein Aiolos (also known as Ikaros family zinc finger protein 3); IRAK1, interleukin 1 receptor associated kinase 1; IRF4, IRF5 and IRF8: interferon regulatory factor 4, 5, and 8, respectively; JAZF1, juxtaposed with another zinc finger protein 1; LBH, LBH regulator of WNT signaling pathway; NFKBIE, nuclear factor of kappa light polypeptide gene enhancer in B-cells inhibitor, epsilon; MED1, mediator of RNA polymerase II transcription subunit 1; MTF1, metal regulatory transcription factor 1; P2RY10, putative P2Y purinoceptor 10; PADI4, peptidyl arginine deiminase 4; PLCL2, phospholipase C like 2; PLD4, phospholipase D family member 4; PPIL4, peptidyl-prolyl cis-trans isomerase-like 4; PRKCH, protein kinase C eta type; PRKCQ, protein kinase C theta; PTPN2, PTNP11 and PTPN22: protein tyrosine phosphatase non-receptor type 2, 11, and 22, respectively; PTPRC, protein tyrosine phosphatase, receptor type C; PVT1, Pvt1 oncogene (non-protein coding); PXK, PX domain containing serine/threonine kinase; RAD51B, DNA repair RAD51 paralog B; RASGRP1, RAS guanyl-releasing protein 1; RCAN1, regulator of calcineurin 1; RTKN2, rhotekin 2; RUNX1, runt-related transcription factor 1; SH2B3, SH2B adapter protein 3; SLC8A3, solute carrier family 8 member A3; SMTNL2, smoothelin like 2; SPRED2, sprouty-related, EVH1 domain-containing protein 2; STAT4, signal transducer and activator of transcription 4; SYNGR1, synaptogyrin 1; TEC, tyrosine-protein kinase Tec; TPD52, tumor protein D52; TNFAIP3, tumor necrosis factor alpha induced protein 3; TNFRSF9 and TNFRSF14: TNF receptor superfamily member 9 and 14, respectively; TRAF1, TNF receptor-associated factor 1; TYK2, non-receptor tyrosine-protein kinase; TXNDC11, thioredoxin domain containing 11; UBASH34, ubiquitin-associated and SH3 domain-containing protein A; UBE2L3, ubiquitin conjugating enzyme E2 L3; YDJC, YdjC chitooligosaccharide deacetylase homolog; ZNF438, zinc finger protein 438.

**Figure 3 jpm-10-00196-f003:**
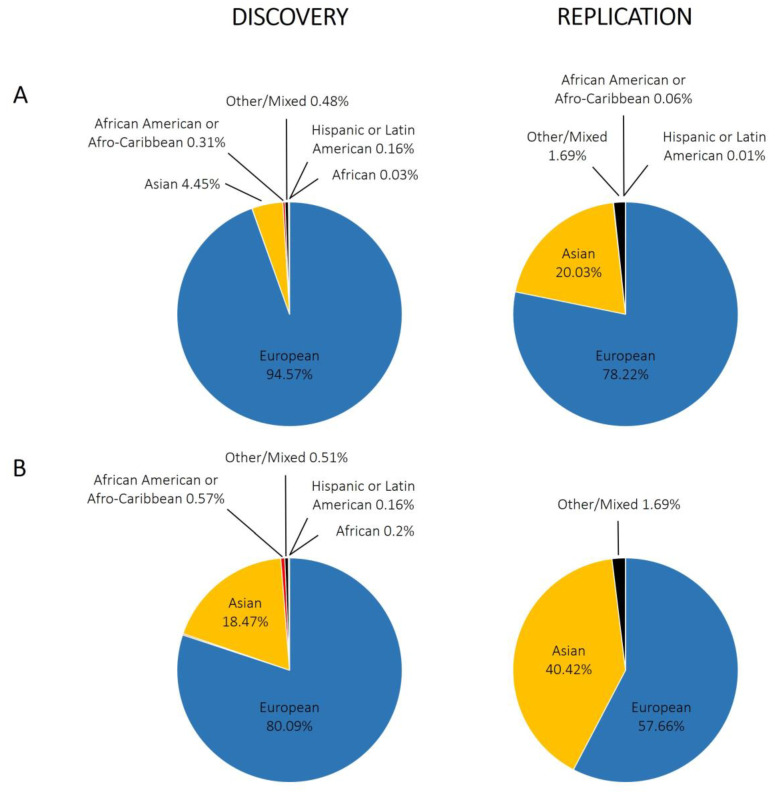
Representation of different ethnic groups in genome-wide association studies performed in the rheumatic disorders. (**A**) Combining different rheumatic disorders: rheumatoid arthritis, systemic lupus erythematosus, osteoarthritis, ankylosing spondylitis, psoriatic arthritis, and Behçet’s disease; (**B**) only rheumatoid arthritis.

**Figure 4 jpm-10-00196-f004:**
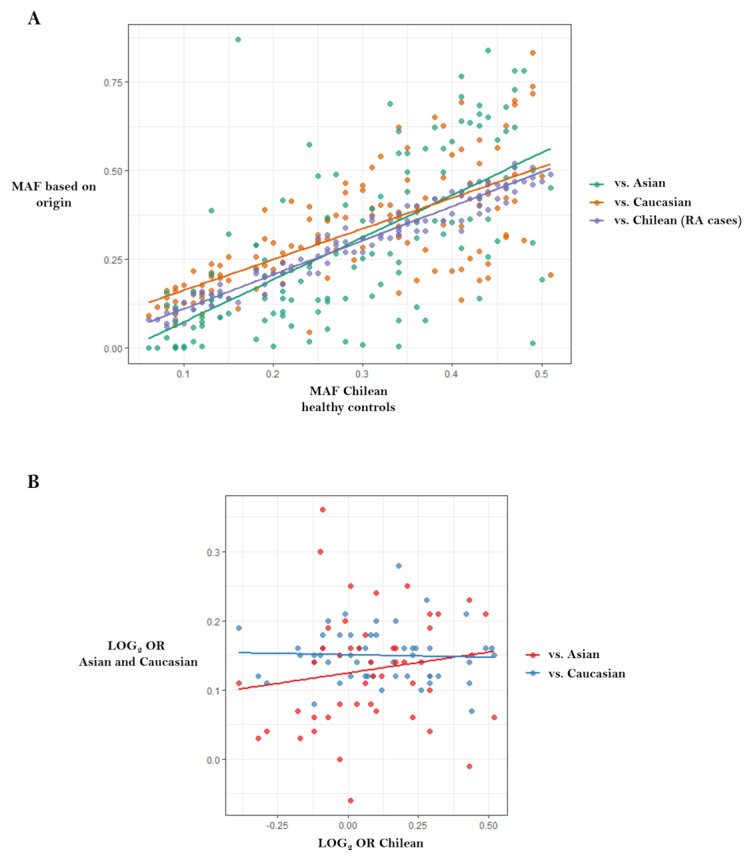
Correlations between data published in genome-wide association studies from Caucasians and Asians versus data reported in the Chilean population. Figure adapted from Castro-Santos et al. [46]: (**A**) Chilean population versus RA Chilean cases, Caucasian and Asian populations (http://www.1000genomes.org) (r = 0.98, r = 0.68 and r = 0.65, respectively); (**B**) Correlation between log (odds ratio) from data published in genome-wide association studies versus log (odds ratio) reported in the Chilean population. There was no correlation between data belonging to Caucasian population and Chileans (r = −0.041, *p* = 0.768), or between Asian populations and Chileans (r = 0.152, *p* = 0.302). Abbreviations: LOG, logarithm; MAF, minor allele frequency; OR, odds ratio.

**Table 1 jpm-10-00196-t001:** Geographic distribution of rheumatoid arthritis-associated human leukocyte antigen (*HLA*)-*DRB1* shared epitope alleles.

Population	HLA-DRB1 Allele(s)	Sequence (70–74 Position)
Caucasoid	* 04:01/* 04:04, * 04:08	QKRAA/QRRAA
Asian	* 04:05	QRRAA
Native Americans	* 14:02	QRRAA
African Americans	* 01:01, * 04:05/* 10:01	QRRAA/RRRAA
Israeli Jews	* 01:01, * 01:02	QRRAA
Latin Americans	* 04:01/* 04:04, * 04:05	QKRAA/QRRAA

*: HLA-DRB1 Allele.

**Table 2 jpm-10-00196-t002:** Basic research studies showing specific changes in DNA methylation in patients with rheumatoid arthritis (RA).

Study	Gene (s)/Region (s) Implicated	Main Results
[59]	Human leukocyte antigen (*HLA*)*-DR-3*	*DR-3* gene was differentially methylated in patients with RA. As a result, the expression of DR-3 protein was downregulated in synovium, thereby providing higher resistance to apoptosis in these cells
[60]	Interleukin 6 *(IL6*)	Lower methylation and subsequent higher expression of IL6 in peripheral blood mononuclear cells in patients with RA
[61]	C-X-C motif chemokine 12 (*CXCL12*)	*CXCL12* gene was hypomethylated in patients with RA and the levels of CXCL12 were subsequently higher in these patients than in those with osteoarthritis, promoting activation of matrix metalloproteinases and joint destruction
[62]	Genome-wide studies comparing stromal fibroblast-like synoviocytes in patients with RA or osteoarthritis	Identification of ~2000 loci differentially methylated, including genes involved in immune response, migration, and cellular adhesion
[63]	Identification of 2 methylation clusters in the major histocompatibility complex (MHC) region associated with epigenetic risk for RA	The DNA methylation study sorted CD14+ monocytes of patients with RA and controls, finding 9 differential methylated sites located in the MHC region and suggesting that monocytes are more proximal to the pathogenic cell type
[64]	Interleukin 6 receptor (*IL6R*), calpain 8 (*CAPN8*), homeobox protein Hox-A11 (*HOXA11*), dipeptidyl-peptidase 4 (*DPP4*), and homeobox protein Hox-C4 (*HOXC4*)	The study showed hypomethylation of *IL6R*, *CAPN8*, and *HOXA11*, and hypermethylation of *DPP4* and *HOXC4*, respectively, in the synovial fibroblasts of patients with RA
[65]	Dual specificity phosphatase 22 *(DUSP22*) and polypeptide N-acetylgalactosaminyltransferase 9 (*GALNT9*)	Multiple sites within *DUSP22* and *GALNT9* genes were consistently hypermethylated and hypomethylated, respectively, in T-lymphocytes from patients with RA
[66]	T-cell surface glycoprotein CD1c (*CD1C*), TNF superfamily member 10 (*TNFSF10*), parvin gamma (*PARVG*), nidogen 1 (*NID1*), dehydrogenase/reductase 12 (*DHRS12*), inositol-tetrakisphosphate 1-kinase (*ITPK1*), acyl-CoA synthetase family member 3 (*ACSF3*), and TNF receptor superfamily member 13C (*TNFRSF13C*)	Although there were differentially methylated genes in patients and control groups, the study showed similar patterns of epigenetic changes in B-lymphocytes from patients with RA or systemic lupus erythematosus
[67]	Poly (ADP-ribose) polymerase family member 9 (*PARP9*)	The study identified an interferon-inducible gene interaction network. The significance of *PARP9* gene methylation and the resulting change in gene expression in the pathogenesis of RA was demonstrated. In addition, the ability of *PARP9* gene to positively regulate interleukin 2, which stimulates various cells of the immune response, was revealed.

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
