# Peer review of "Latin American Genes: The Great Forgotten in Rheumatoid Arthritis"

_jpm, 2020, doi:10.3390/jpm10040196_

Round 1

Reviewer 1 Report

The paper entitled "Latin American genes: the great forgotten in rheumatoid arthritis" reviews the influence of the latin american genes in rheumatoid arthritis.

The present review is well organized and presented. In my opinion the paper is suitable to be published in Journal of Personalized Medicine after minor revisisons.

Comments:

  • The abbreviations should be described when used for the first time
  • "Type of the Paper: Review" should be "Review"
  • The Figures caption is below the figure itself, not above. Please correct.

Author Response

  1. The abbreviations should be described when used for the first time

R: We have now specified what each abbreviation (particularly, but not only, gene names) stands for throughout the entire manuscript (including Tables and table footnotes). In addition, we have now avoided the use of abbreviations for those that only appeared once in the manuscript. Please see revised manuscript.

  1. "Type of the Paper: Review" should be "Review"

R: Done.

  1. The Figures caption is below the figure itself, not above. Please correct.

R: Amended.

Thank you, on behalf of all coauthors.

Reviewer 2 Report

This is a narrative review about the role of genetic determinants in rheumatoid arthritis risks, with a focus on the Latin American population. The paper is interesting and well written. However I have the following comments for the authors:

  • Please correct the incorrect lines of the pie charts in Figure 2
  • Page 9, line 380. Please specify what the “[…] visual analog scale […]” refer to.
  • Please consider to add one or more section about non-HLA genes (eg: cytokines and their receptors, chemokines and their receptors, costimulatory molecules and intracellular pathways) and the risk of rheumatoid arthritis
  • Please consider to extend the Pharmacogenetics section with data regarding bDMARDs response.
  • Page 10, line 420 “[…] mRNA […]” is it miRNA?

Author Response

  1. Please correct the incorrect lines of the pie charts in Figure 2

R: Done. Please note that former Figure 2 has now become Figure 3.

  1. Page 9, line 380. Please specify what the “[…] visual analog scale […]” refer to.

R: Visual analog scale, VAS, is a psychometric response scale for parameters that range across a continuum of values, such as pain. The VAS pain scale ranges from “no pain” to “worst pain,” and patients mark a line to indicate how they are feeling. We have added two references to clarify the outcome measures, the tender joint count (TJC) and VAS (Page 10, line 470; reference 54 and 55, respectively).

  1. Please consider to add one or more section about non-HLA genes (eg: cytokines and their receptors, chemokines and their receptors, costimulatory molecules and intracellular pathways) and the risk of rheumatoid arthritis

R: Thank you for the comment. According to the Reviewer’s suggestion, we have now included more information regarding non-HLA genes and the risk of rheumatoid arthritis (Section 3.2.1. Non-HLA genetic associations; and Figure 2). Please see revised manuscript, pages 4-6, lines 167-247.

  1. Please consider to extend the Pharmacogenetics section with data regarding bDMARDs response.

R: We have now included more information regarding bDMARDs response in the Pharmacogenetics section. Please see revised manuscript, pages 10, lines 472-483 and page 12, lines 526-528.

  1. Page 10, line 420 “[…] mRNA […]” is it miRNA?

R: No. We meant to say that methylation is stable over time, although it can be modified, compared to the fluctuating nature of protein or messenger RNA.

Thank you, on behalf of all coauthors.

Round 2

Reviewer 2 Report

I have only one further comment for the authors: the fourth component of DAS28 is general health assessment on a VAS (10.1002/art.1780380107), not pain.